# Disruption of Sex-Linked *Sox3* Causes ZW Female-to-Male Sex Reversal in the Japanese Frog *Glandirana rugosa*

**DOI:** 10.3390/biom14121566

**Published:** 2024-12-09

**Authors:** Ikuo Miura, Yoshinori Hasegawa, Michihiko Ito, Tariq Ezaz, Mitsuaki Ogata

**Affiliations:** 1Amphibian Research Center, Hiroshima University, Higashi-Hiroshima 739-8526, Japan; 2Institute for Applied Ecology, University of Canberra, Bruce, ACT 2617, Australia; tariq.ezaz@canberra.edu.au; 3Kazusa DNA Research Institute, Kisarazu 292-0818, Japan; yhasega@kazusa.or.jp; 4School of Science, Kitasato University, Sagamihara 252-0373, Japan; michichikoito.ito@gmail.com; 5Preservation and Research Center, City of Yokohama, Yokohama 241-0804, Japan; zvp06246@nifty.com

**Keywords:** female determination, female heterogamety, loss of function, amphibia

## Abstract

*Sox3* is an ancestral homologous gene of the male-determining *Sry* in eutherian mammals and determines maleness in medaka fish. In the Japanese frog, *Glandirana rugosa*, *Sox3* is located on the Z and W chromosomes. To assess the sex-determining function of *Sox3* in this frog, we investigated its expression in gonads during early tadpole development and conducted genome-editing experiments. We found that the *Sox3* mRNA levels in the gonads/mesonephroi were much higher in ZW females than that in ZZ males, and that the W-borne allele was dominantly expressed. A higher expression in ZW females preceded the onset of the sexually dimorphic expression of other autosomal sex differentiation genes. The Sox3 protein was detected by immunostaining in the somatic cells of early tadpole gonads around the boundary between the medulla and cortex in ZW females, whereas it was outside the gonads in ZZ males. Disrupting *Sox3* using TALEN, which targets two distinct sites, generated sex-reversed ZW males and hermaphrodites, whereas no sex reversal was observed in ZZ males. These results suggest that the sex-linked *Sox3* is involved in female determination in the ZZ-ZW sex-determining system of the frog, an exact opposite function to the male determination of medaka *Sox3y* and eutherian *Sry*.

## 1. Introduction

Genetic sex determination is labile in vertebrates [1,2,3,4]; there are multiple types of sex-determining genes. Since the discovery of the male-determining gene *Sry* in humans, 17 sex-determining genes have been identified to date [5,6,7]. These include transcription factors, cytokines, hormones and their receptors, enzymes catalyzing steroid hormones, and even proteins that are not related to sex. Contrary to the XX-XY system, fewer sex-determining genes have been identified in the ZZ-ZW system. They include *Dmrt1*, showing dose-dependent sex determination in chickens and flatfish [8,9,10], the dominant female-determining gene *dm-W* (a partial duplicate of *Dmrt1*) in clawed frogs [11], and *Sox2* and *nr5a1*, which are candidates in turbot fish and dragon lizards, respectively [12,13]. The molecular mechanisms of female determination in the ZZ-ZW system remain largely unclear, in contrast to the XX-XY mechanisms, and the transition mechanisms from male to female heterogamety are far from determined.

The Japanese frog *Glandirana rugosa* is unique, as two distinct systems, XX-XY and ZZ-ZW, are present within the species [14]. These are separated by geographic population. It is estimated that the ZZ-ZW system originated 4.8 million years ago, based on mitochondrial DNA [15,16], from hybridization between the two ancestral populations of West Japan and the sister species *G. reliquia* (the former East Japan of *G. rugosa*), both of which have XX-XY systems with non-homologous sex chromosomes [14]. Therefore, this frog is a suitable model for understanding the transition mechanism of sex determination from XX-XY to ZZ-ZW, that is, the evolutionary mechanism of female heterogametic sex determination. At present, the sex-linked androgen receptor gene (*Ar*) is thought to be the sex-determining gene in the ZZ-ZW system of *G. rugosa* [17,18]. However, the functional analyses carried out during the studies are questionable, because the sex-reversed tadpoles that were used to evaluate the gain and loss of gene function were obtained by exposing them to testosterone.

*Sox3* encodes a transcription factor and is the ancestral gene of the eutherian male-determining gene, *Sry* [19,20,21,22]. In the medaka fish, *Sox3* on the Y chromosome has evolved as a male-determining gene [23,24]. In *G. rugosa*, *Sox3* is located on the Z and W chromosomes [25]: the Z- and W-borne *Sox3* show 99.1% homology in the nucleotide sequence, encoding an identical amino acid sequence [15]. *Sox3* is highly expressed in the ZW gonads of tadpoles [26]. Although this gene is a strong candidate for determining femaleness in the frog, no functional analyses have been performed. In the present study, we conducted expression and functional analyses of *Sox3* to elucidate its role in sex determination. We also verified the sex-determining function of the *Ar* gene using transgenesis.

## 2. Materials and Methods

### 2.1. Frogs

The Japanese frog *Glandirana rugosa* used in this study belongs to the ZW population (North-west Japan) and has ZZ-ZW-type heteromorphic sex chromosomes [14]. The frogs were collected from Nagaoka city, Niigata Prefecture, Japan, and thus belonged to the ZW1 sub-population [27]. The sex of the specimens was determined via the inspection of their gonads after anesthesia with 0.03% Benzocaine. The genetic sex was determined based on the genotypes of two sex-linked genes, ADP/ATP translocase and steroidogenic factor 1, according to previously described genotyping protocols [27].

### 2.2. Gene Expression Analysis

The gonads, attached to mesonephros, were cut out from ten male and ten female tadpoles at each stage, after anesthesia with 0.01% MS222, and they were quickly immersed and pooled in different tubes with RNAlater (Thermo Fisher Scientific Inc., Waltham, MA, USA); the gonads were separated by genetic males and females, incubated for one night at 4 °C and were finally stored at −80 °C until use. RNA extraction, cDNA synthesis, and the qualitative real-time PCR of six genes, namely *Sox3*, *Ar*, *Cyp17*, *Dmrt1*, *Cyp19* and *Foxl2*, were performed according to the method of Miura et al. (2016) [28]. The same primer sets were used: Sox3-forward, 5′-CAACAGATGCACAGGTACGACA-3′ (+573) and reverse, 5′-GCAGGTGGGGGAGAACTAGG-3′ (+734). The β-actin gene was used as the reference gene to calculate the relative expression rate of the target gene.

### 2.3. Histochemical Staining with Anti-Frog Sox3 Antibody

#### 2.3.1. Preparation of Antibody and Specificity Analysis

The synthetic polypeptide of eighteen amino acids (GDASDPSSLQSRLHSVH, +270~+287)) of *G. rugosa* SOX3 conjugated with KLH, which has low homology to *G. rugosa* SOX2, was used with immune rabbit antibody to generate anti-GrSOX3 polyclonal antibodies according to the instructions of Sigma Life Science (Merck KGaA, Darmstadt, Germany). The antibody specificity of the anti-GrSOX3 antibody was confirmed by Western blot analysis. The 2 μg construct DNA, including *Sox3* of *G. rugosa* under the promoter of *Xenopus laevis* elongation factor, was introduced into cultured Chinese hamster cells (3 cm dish) with 7 μL of FuGene HD transfection reagent (Promega, Madison, WI, USA). Then, 24 h after transfection, proteins were extracted from the transfected cells in RIPA buffer (Nakalai Tesque, Kyoto, Japan), which included the protease inhibitor cocktail, and the final extracts were boiled (5 min at 95 °C) and separated on 12% SDS-PAGE gel with 1x TG (Tris-Glycine) buffer; they were then transferred onto a polyvinylidene fluoride (PDVF) membrane. Western blotting was carried out using the anti-GrSOX3 antibody (1:1000) and alkaline phosphatase-conjugated goat anti-rabbit IgG (1:1000; Cell Signaling Technology, Beverly, MA, USA); NBT was used for staining (Appendix A).

#### 2.3.2. Whole Mount Immunostaining

The body parts of the tadpoles, including the gonads and mesonephros, were cut and fixed with 4% PFA in PBS on ice for one night. Then, the tissue was washed with PBS for 5 min three times and incubated with 6% H_2_O_2_ in methanol for 1 h at room temperature, which was then replaced with fresh methanol. The tissue was moved to a solution of methyl alcohol/DMSO (1:1) containing 10% Triton and incubated for 30 min on ice, and then incubated in 1x TST buffer (Tris, NaCl, Triton X100, pH 7.8); the buffer was changed three times, each for 5 min, on ice. The tissue was incubated (blocking) in 5% skim milk dissolved in 1x TST for one night at 4 °C, and then incubated with anti-GrSOX3 antibody (1:1000) in 1x TSTM containing 5% skim milk for one night at 4 °C. After washing with 1x TST 6 times, each for 6 min, the tissue was incubated with anti-rabbit IgG AP-linked antibody (1:1000, Cell Signaling Technology) for one night at 4 °C, and washed 6 times with 1x TST, each for 10 min. Finally, the tissue was stained in alkaline phosphatase (AP) buffer containing NBT. After being rinsed with 1x TST, 1x TST/Ethanol for 5 min and Ethanol for 1 h, the tissue was stored in 1x TST and observed, or was forwarded for serial transversal sectioning.

#### 2.3.3. Transversal Section Immunostaining

The gonad–mesonephros complex of the tadpoles was cut out and fixed with 4% PFA in PBST (PBS including 0.1% Triton X) on ice for one night. Then, the tissue was gradually dehydrated with ethanol from 25% to 75% and finally replaced with ethanol, and embedded with paraffin after being passed through Xylene. Transversal sectioning was carried out, and the sections (10 μm in thickness) were hydrated on a slide glass after removing the paraffin passed through Xylene; the sections were then incubated in 1x TBST. Then, after being incubated in blocking solution (1:1000, 5% normal goat serum in TBST) for 1 h at room temperature, the sections were incubated with anti-GrSOX3 antibody (1:1000, 5% goat serum in TBST) for one night at 4 °C. After being washed with TBST three times, each for 5 min at room temperature, the sections were incubated with anti-rabbit IgG-FITC (1:1000, 5% normal goat serum in TBST (Thermo Fisher Tech., Waltham, MA, USA) for 1 h at room temperature in a dark box. The sections were washed in TBST three times, each for 5 min, stained with Propidium iodide (PI) for 10 min, and covered with Vectashield mounting medium (VECTOR LABORATORIES, Inc., Newark, CA, USA). The sections were observed with a fluorescence microscope (Nikon Eclipse 80i; Nikon Inc., Tokyo, Japan) and photographed (Nikon DS camera control unit DS-U3; Nikon DS camera control unit DS-U3).

### 2.4. TALEN Knockout and Transgenesis

#### 2.4.1. Sox3 TALEN Constructs

Three sets of TALEN constructs based on pTAL.T7 vectors [29] targeting three distinct sequences (A, B and C) of *Sox3* of *G. rugosa* were designed and prepared; these were obtained from Cellectis Bioresearch (Paris, France) (Figure 1). mRNA was transcribed from 1 μg of each construct according to the instructions of the mMessage mMachine kit (Thermo Fisher, Scientific KK, Tokyo, Japan).

#### 2.4.2. Ar Transgenesis Construct

The androgen receptor cDNA fragment (2675 bp) of *G. rugosa*, whose mRNA was extracted from the ZZ male tadpoles, was amplified using the primer set (forward: 5′-CCTATCCCTTTGCTACCCTCAG-3′, reverse: 5′-TGCCTTTCTGATATGACACGTTGA-3′), and was replaced with the EGFP fragment downstream of the *Xenopus* elongation factor promoter (xEF) in p2AL200R150G to prepare the xEF-Ar fragment. Then, the two fragments of xEF1-Ar and xEF1-EGFP were tandemly connected and inserted into the I-Sce-I site of pCMV-GFP-Sce after removing the original *Sal*I-*Not*I fragments, including CMV-GFP (Appendix A). The constructs were purified using the FastGene™ Gel/PCR Extraction Kit (Nippon Genetics, Tokyo, Japan).

#### 2.4.3. Microinjection

The females of *G. rugosa* were ovulated by the injection of LHRH (Salmon, 4013835, Bachem AG, Bubendorf, Switzerland) dissolved in Holtfreter’s solution (Holt), according to the method of Berger et al. (1994) [30], and the eggs were artificially inseminated with sperm from the males according to the method of Ohtani et al. (2003) [31]. Then, 20 min after insemination, the fertilized eggs were de-jellied with 2% Cystein in 0.1x MBC buffer (pH 8.0) [32] for 8–10 min at room temperature by changing the solution twice to remove the jelly completely. The de-jellied eggs were washed eight times with 1x Holt solution and were finally immersed in 6% Ficoll dissolved with 1x Holt. The eggs were placed on a slide glass and microinjected with DNA (transgenesis) or RNA (TALEN knock out) using an injector (Nanoject II, Drummond, Broomall, PA, USA). After finishing the injection, the eggs were moved to new 6% Ficoll solution and incubated for 2 h at 16–18 °C. From the eggs returned to room temperature, normally developing embryos (at 2 to 4 cell cleavage stage) were selected and incubated in 0.5x Holt with gentamycin (25 μg/mL) for two nights, in 0.1x Holt with gentamycin (5 μg/mL) for one night and finally returned to dechlorinated tap water. For TALEN knockout, 1.25 μg of mRNA transcribed from construct A (right arm) and 1.25 μg of mRNA from construct B (left arm), both targeting the *Sox3* sequence of *G. rugosa*, were mixed and dissolved in 23 μL of distilled water. One drop (4.6 μL, including 500 pg of RNA) was injected into an egg. For transgenesis, we used the method of meganuclease I-*Sce*-I [32]. The 500 ng–1 μg DNA of the I-Sce-I construct, including the androgen receptor gene of *G. rugosa* and xEF1-EGFP (Appendix A), were digested with I-*Sce*-I (25 unites) for 40 min at 37 °C and then put on ice until they were used for microinjection.

#### 2.4.4. Identification of Sox3 Mutations

The whole or a part of the gonads was cut out from the ZW sex-reversed frogs and used for identifying the *Sox3* mutations. Genomic DNA was extracted from the gonad using NucleoSpin tissue (TaKaRa Bio. Inc., Kyoto, Japan) or the Kaneka easy DNA extraction kit version 2 (Kaneka Corp., Tokyo, Japan), according to the manufacturer’s instructions. On the other hand, RNA extraction and cDNA synthesis were carried out by the same method used in the gene expression analysis of Miura et al. (2016) [28]. *Sox3* fragments, including the TALEN target sites, were amplified from DNA or cDNA using the following primer set: forward (−62~) 5′-GTGCGCTCCTCCTGCTTCTTT-3′ and reverse-1(~+991) 5′-TCCTCAAGTTTTCTGCATTCTGAT-3′ or reverse-2 (~+878) 5′-TGCACTTTGGTAATGTTGGTGG-3′. These fragment were then purified using the FastGene™ Gel/PCR Extraction Kit (Nippon Genetics, Tokyo, Japan). The cloning of the purified fragments was carried out using the Mighty cloning kit (Takara) and ECOS competent *E. coli* (Nippon Gene Co., Ltd., Tokyo, Japan) into the DH5 α bacterial strain. Twenty to thirty positive clones were selected and the DNA from each clone was extracted using the conventional method [33]. After purification, the nucleotide sequence of the insert of each clone DNA was determined using an ABI PRISM 3130xl genetic analyzer (Applied Biosystems, Waltham, MA, USA, Thermo Fishers), according to the manufacturer’s instruction.

## 3. Results

### 3.1. W-Borne Sox3 Is Dominantly Expressed in ZW Gonad of Early Tadpole

To unravel the differences in the expression profile of the gonads between sex-linked *Sox3* and other autosomal sex-differentiation genes, we investigated the mRNA expression levels in the gonad/mesonephros complexes of ZZ males and ZW females during early tadpole development from 12 to 23 days post-fertilization (dpf) using q-PCR. *Cyp17* and *Cyp19* are involved in the synthesis of testosterone and estradiol, respectively, and *Dmrt1* and *Foxl2* are involved in the differentiation of the testes and ovaries in vertebrates, respectively. At early tadpole stages, the expressions of the four autosomal genes swung between males and females, probably due to individual variations (ten males and ten females were used for the one-stage analysis), but after the fixed stages, their sexually dimorphic expressions became stable; here, these are designated as the onset stage of sexually dimorphic expression. They were 14 dpf (*Cyp19*) and 15 dpf (*Foxl2*) in ZW females and 15 dpf (*Cyp17*) and 23 dpf or later (*Dmrt1*) in ZZ males (Figure 2). On the other hand, the *Sox3* expression was much higher consecutively in ZW females than in ZZ males from 12 dpf to 23 dpf and preceded the onset of the sexually dimorphic expression of other autosomal sex-differentiation genes. In addition, W-borne *Sox3* was predominantly expressed from 12 to 14 dpf and contributed to the higher expression in ZW females (Figure 3).

### 3.2. Sox3 Is Expressed in Somatic Cells of ZW Gonad

To identify the gonadal cells expressing Sox3 during early tadpole development, we investigated protein localization by histochemical immunostaining using an anti-frog Sox3 antibody (Appendix A). A positive signal was detected in the somatic cells of the gonads of ZZ males and ZW females at 18 dpf and it was located at the medulla or at the boundary between the medulla and cortex in ZW females; however, it was outside the gonads of ZZ males (Figure 4). In addition, we observed that the Sox3-positive cells migrated from the mesonephros into the ZW bipotential gonad (Appendix A).

### 3.3. Disruption of Sox3 Causes ZW Female-to-Male Sex Reversal

To assess the sex-determining function of *Sox3* in the frog, we disrupted the genomic *Sox3* gene using TALEN by targeting three distinct sequences. Two of them (A and B) were located between the starting codon ATG and the DNA-binding (HMG) domain, whereas the other (C) was located within the transcription activation (TA) domain (Figure 1). To evaluate complete sex reversal, we reared the experimental frogs for up to two years after fertilization, over metamorphosis until almost complete sexual maturation. Of the 15 frogs, in which *Sox3* sequence A was targeted, two ZW frogs were hermaphrodites with both testes and ovaries, whereas the other ZW frogs were females (Figure 5A,B). Of the 46 frogs, in which sequence C was targeted, one ZW was a hermaphrodite, two ZW were males, and the other ZW frogs were females (Figure 5C and Figure 6A,B). Of the 19 frogs, in which both sequences A and C were targeted, one ZW frog was male and the other ZW frogs were females (Figure 6C). No ZW sex-reversed frogs were obtained by targeting sequences B, or B and C. All ZZ frogs in the experiments were males with testes (Table 1 and Table 2).

The testes of ZW males and hermaphrodites were histologically examined and were found to be composed of testicular tubules, including meiotic cells or spermatids (Figure 5 and Figure 6), and those of two males (1C-ZW1 and 3C-ZW1) contained sperm bundles (Figure 5C and Figure 6B). When the 3C-ZW1 male was crossed with six wild-type ZW females, viable offspring were obtained, indicating normal fertility (34% fertility of the control male) (Appendix A). In addition, 26% of the offspring were WW homozygous for the sex-linked *SF1* and died of edema at 10 dpf due to the typical lethal WW symptoms of the frog (Appendix A) [34], proving that the sex-reversed male was a ZW frog with regard to its sex chromosomes.

To confirm the disruption of the *Sox3* gene, we sequenced genomic *Sox3* genes and cDNA isolated from the gonads of two ZW males and three ZW hermaphrodites. Various types of mutations, such as deletions, insertions, and substitutions, were identified (Appendix A): 14.8–94.1% mutations in ZW testes and 21.4–84.6% mutations in ZW ovaries were identified (Table 3).

### 3.4. Change of Gene Expressions in ZW Testes by Disrupting Sox3

To confirm ZW male sex reversal, we investigated the expression of the male-specific genes *Cyp17* and *Dmrt1* and the female-specific gene *Cyp19* in their gonads. In four of the five ZW testes examined, the expression of both *Cyp17* and *Dmrt1* or *Dmrt1* was much higher than that in the ovaries of the control ZW female, whereas they were low in the testis of 2AC-ZW1. Sperm bundles were observed in the testes of two ZW males (1C-ZW1 and 3C-ZW1), in which both *Cyp17* and *Dmrt1* were upregulated (Figure 7). In all ZW testes, *Cyp19* expression was either very low or zero. In contrast, in the ovaries of the ZW hermaphrodites, the expression of both *Cyp17* and *Dmrt1* was as low as in normal ovaries, whereas *Cyp19* expression was also low or zero in two ovaries (left ovary of 1A-ZW1 and right ovary of 1C-ZW1) and slightly higher in the other ovary (left ovary of 1C-ZW1) (Figure 7).

### 3.5. ZW Female Is Not Sex-Reversed by Up-Regulating Sex-Linked Androgen Receptor Gene

The androgen receptor gene (*Ar*) is located on the Z and W chromosomes in the frog *G. rugosa* and the W-borne allele is very faintly expressed, indicating a system of double dose in ZZ males and a single dose in ZW females. The *Ar* gene is postulated to be involved in sex determination in the frog [17,18], as a mimic of *Dmrt1* dose-dependent sex determination in the ZZ-ZW system of chickens. We confirmed in previous and present studies that *Ar* expression is higher in ZZ than that in ZW gonad–mesonephros complexes (Figure 2B) [28] at the early tadpole stage, and that it precedes the onset of the sexually dimorphic expression of other sex differentiation genes, as *Sox3* does. However, our results regarding ZW male sex reversal induced by *Sox3* disruption indicate that a single dose of *Ar* gene is sufficient to determine maleness; thus, a double dose is not always necessary. To assess the male-determining function of the *Ar* gene, we generated *Ar*-transgenic frogs by introducing a construct containing frog *Ar* under the control of *Xenopus* elongation factor promoter (xEF1), which was expected to be expressed ubiquitously, into fertilized eggs of the frog species (Appendix A). We identified founders by detecting EGFP fluorescence because xEF1-EGFP was included in the same construct together (Appendix A). In addition, we confirmed the elevation of *Ar* expression in the muscles and gonads of ZW transgenic frogs, which was almost two-fold higher than that in ZW frogs. The 14 ZW F1 and 20 F2-ZW offspring of the *Ar*-transgenic frogs did not show any ZW male sex reversal (Figure 8; Appendix A), indicating that the upregulation of the *Ar* gene, that is, a double dose of *Ar*, does not determine maleness in the frog.

## 4. Discussion

### 4.1. Sex-Linked Sox3 Determines Femaleness in G. rugosa

The disruption of sex-linked *Sox3* in the frog *Glandirana rugosa* produced ZW sex-reversed males and hermaphrodites but caused no sex reversal in ZZ males. One ZW male (3C-ZW1) was fertile and produced the next generation. The male-specific genes *Cyp17* and *Dmrt1* were upregulated in ZW testes, whereas the female-specific gene *Cyp19* was downregulated, even in the ovaries of ZW hermaphrodites, where high rates of *Sox3* mutations were identified, suggesting that *Cyp19* expression is under the control of *Sox3*, as proposed by Oshima et al. (2009) [26]. These results imply that sex-linked *Sox3* is necessary for determining femaleness in the frog, an exactly opposite function to the male determination of *Sox3y* in medaka fish and *Sry*, which evolved from *Sox3* in eutherians. The mutation rates ranged from 14.8 to 94.1% in the ZW testes and were also high in the ovaries of ZW hermaphrodites. Despite the high mutation rates acquired, the ovaries of ZW hermaphrodites did not differentiate into testes possibly because the mutations were skewed to germ cells. For example, in 3C-ZW1 males, the mutation rate in genomic *Sox3* was 44.2%, whereas it was only 14.8%, which is very low, in *Sox3* cDNA, and they were all in-frame mutations (three base deletions) (Table 3; Appendix A). Mutations in cDNA, but not in genomic DNA, were found to be transmitted to the next F1 generation, showing that in the ZW testes, mutations occurred very slightly in germ cells but heavily in somatic cells. This implies that *Sox3* expression in the somatic cells of the gonad is crucially important for female determination. Another reason may be that the mutations were skewed to the Z allele, though not to the W allele in hermaphrodite ovaries (we could not discriminate between the Z and W mutations because no allele-specific primers were available for *Sox3* amplification). The upregulation of *Sox3* from the W allele may be important for female determination because it is dominantly expressed and contributes to a higher expression in ZW tadpole gonads from the extremely early stages of tadpole development. The acquisition of the unique expression of W-borne *Sox3* may have been crucial for the birth of the female-determining gene and ZZ-ZW system in the frog. We hypothesize that the *Sox3* gene on the original chromosome 7 acquired an enhancer element to induce its higher and ectopic expression in the somatic cells of the gonads during the evolution of the original population.

### 4.2. Sex-Hormone Independent Sex-Determination in the Frog ZZ-ZW System

In the testes of three ZW males and two hermaphrodites, the expression of *Cyp17* (involved in testosterone synthesis) was very low; however, typical seminal tubules were formed, suggesting that testis determination in this frog was independent of sex hormones. Similarly, in the ovaries of ZW hermaphrodites, the expression of *Cyp19* (involved in estrogen synthesis) was extremely low; however, typical ovaries were formed, including large auxocytes, suggesting again that female determination is independent of sex steroid hormones, as proposed by Miura et al. (2016) [28], who showed that the aromatase (*Cyp19*) inhibitor did not induce any ZW male sex reversal in this frog. In the ZW male frogs and hermaphrodites (1C-ZW1 and 3CZW1), sperm bundles were observed in the testes, where both *Cypt17* and *Dmrt1* were upregulated; however, in the testes without sperm bundles, neither or only *Dmrt1* were upregulated (1A-ZW2, 1C-ZW2, and 2AC-ZW1) (Figure 7). Therefore, steroid hormones and the expression of *Sox3* in germ cells may be important for gametogenesis but not for sex determination, as proved in *Sox3* null mice [35]. During the evolution of the ZZ-ZW system in the frog, sex determination mechanisms evolved from sex steroid sensitivity to resistance, in association with sex chromosome heteromorphy [28]. The results of the present study support the evolutionary scenario of sex determination in *G. rugosa*.

### 4.3. How Does Sox3 Determine an Ovary Differentiation in ZW Bipotential Gonad?

The male-determining gene *Sry* of eutherians, which evolved from *Sox3*, is first expressed in somatic (Sertoli) cells of the XY bipotential gonad and determines its future as a testis. Even the homologous *Sox3* on the X chromosome, if ectopically expressed in the Sertoli cells, can determine testis differentiation [36]. At the next stage of testis differentiation, the mesonephric (epithelial) cells migrate into the XY gonad to form testicular cords [37,38]. Then, how does the *W-Sox3*, which is similarly expressed in the somatic cells of the bipotential gonad, determine an ovary in the frog *G. rugosa*? The function of *Sox3* is completely opposite to the male determination in eutherians and medaka fish. Our hypothesis is that *W-Sox3* is first expressed highly in mesonephros and then migrates into the ZW bipotential gonad, positioning itself at the center or boundary between the medulla and cortex (Figure 4, Figure 9 and Appendix A). The Sox3-expressing mesonephric cells separate the medulla from the cortex to inhibit the movement of germ cells from the cortex and form an ovarian cavity. Together, *W-Sox3* may promote the up-regulation of *Cyp19* in the cells, as suggested by Oshima et al. (2009) [26], to synthesize estradiol and contribute to the development and growth of the ovaries. Hereafter, it is necessary to identify the direct target gene and the downstream genes of *Sox3* to confirm our hypothesis.

### 4.4. Sex-Linked Androgen Receptor Gene Is Not Involved in Sex Determination

The 34 *Ar*-transgenic ZW frogs (F1 and F2) we experimentally produced were all females with ovaries, and not hermaphrodites or males, indicating that the *Ar* gene is not involved in sex determination. In addition, the ZW males and hermaphrodites produced by *Sox3* disruption indicate that a single dose of *Ar* gene on the Z chromosome is sufficient to induce maleness; however, a double dose is not always necessary. In contrast, the resistance to sex reversal in ZW *Ar*-Tg frogs suggests that a dominant female determiner exists on the W chromosome. Our results suggest that Sox3 is the female determiner.

Ohtani et al. (2003) [31] found that the *Ar* gene that is located on the W chromosome is not expressed or has a very low level of expression in *G. rugosa* using WW tadpoles. Then, *Ar* dose-dependent sex determination in the frog, with a double dose to males and a single dose to females, was proposed by Yokoyama et al. (2009) [39]. Recently, gain- and loss-of-function analyses have been conducted to prove the sex-determining function of the *Ar* gene in the frog [17,18]. However, their functional analyses are complicated: no sex reversal occurred in the transgenic ZW frogs and the knockdown (KD) of ZZ provided no results; instead, they administered testosterone to the transgenic and KD ZW tadpoles to induce sex reversal or the elevation of *Cyp17* expression in the gonads. These experiments are far from clear evidence of the male-determining function of *Ar* because it is evident that the administration of testosterone to tadpoles induces the elevation of *Cyp17* in gonads [28] and that, at higher concentrations, genetic ZW is sex reversed into a phenotypic male [34,40]. In addition, the fact that absence of (or faint) *Ar* expression from the W allele is not common in the ZW populations of *G. rugosa* refutes the *Ar* sex-determining function. The ZW populations are divided into three geographic populations, ZW1, 2, and 3, based on their mitochondrial DNA. The *Ar^w^* of ZW females from the ZW1 and 2 populations is degenerated and shows almost no expression. In contrast, the *Ar^w^* expression in ZW females from the ZW3 population depends on the female, as they are active in some but inactive in others [27]. In addition, the *Ar^w^* of the ZW females in the Neo-ZW population (another female heterogametic population) is as active in expression as that of *Ar^Z^* [27]. This implies that *Ar* dose-dependent sex determination is not common in ZW systems; however, another factor determining femaleness should exist.

### 4.5. Molecular Mechanisms of Female Heterogametic Sex Determination

In contrast to male heterogametic sex determination (XX-XY), fewer sex-determining genes have been identified or suggested in other animals: one in birds, one in reptiles, one in frogs and three in fish [5,6,7]. Only in two genes has sex determination been proven by functional analyses. These are *Dmrt1* and its derivative. In chickens, ZZ males receive a double dose of *Dmrt1* to determine maleness, whereas ZW females receive a single dose of *Dmrt1* to determine femaleness. The up-regulation of *Dmrt1* causes sex reversal in ZW males, whereas the down-regulation of *Dmrt1* causes sex reversal in ZZ [8,9]. A similar case has been reported in flatfish [10]. The *dm-W* in a tetraploid clawed frog, a partial duplicate of *Dmrt1* in the *S* genome, determines femaleness by dominantly inhibiting the male-determining function of *Dmrt1* [11]. In *Seriola* fish, a steroidogenic enzyme (HSD17b1) determines femaleness; only one amino acid of the enzyme catalyzing the interconversion between testosterone and estradiol is different between the Z- and W-borne alleles. The 144th glutamic acid residue of the Z chromosome (Z^E^) is responsible for the attenuation of enzyme activity, whereas glycine (W^G^) recovers enzyme activity to produce estradiol and determines femaleness [41]. In the dragon lizard, *nr5a1*, a nuclear receptor subfamily, is located on the Z and W chromosomes, and multiple isoforms are transcribed from the Z- and W-borne alleles [8]. It has been hypothesized that truncated isoforms transcribed from the W-borne allele inhibit the formation of full-length intact proteins and suppress testis determination. Except for *HSD17b1* in Seriola fish, ovary determination is accomplished by inhibiting testis determination.

Sex-linked *Sox3* in the frog *G. rugosa* is highly expressed in the somatic cells of the ZW gonads of early tadpoles, inside the medulla, or at the boundary of the cortex. As for female determination in the frog, we hypothesize that *Sox3*-expressing mesonephric cells form the ovarian cavity by inhibiting the migration of germ cells from the cortex. A strong candidate for the female-determining gene in turbot fish is *Sox2*, a sister protein of Sox3, which is expressed in neural crest cells at the neural stage of embryo development and is involved in neurogenesis [12,42]. In addition, a commonality in the evolutionary histories of the frog and fish is that both ZZ-ZW systems have recently been derived from other XX-XY systems [12,14]. *Sox2ot* (long non-coding RNA) has been hypothesized to regulate the expression of *Sox2*, which is located within an intron of *Sox2ot* in fish [43,44]. Such non-coding RNA might have been borne around the W-borne *Sox3* region in the frog and changed the original expression profile of sex-linked *Sox3*. Or some regulatory connection between *Sox3* and *Sox2* could exist in the sex-determining cascade, as has been shown in neurogenesis [45].

Since the discovery of *Sry*, four conditions must be satisfied to prove the sex-determining function of a candidate gene: (1) its genomic location on sex chromosomes, (2) its sexually dimorphic expression in gonads, (3) the sex reversal caused by the gain of function, and (4) the loss of function. The third condition is not yet satisfied in *Sox3* of *G. rugosa*. For that, the crucial enhancer that regulates *Sox3* expression in the somatic cells of ZW tadpole gonads must be identified. We will test this approach using sequence information from the whole genome of *G. rugosa*.

## 5. Conclusions

*Sox3* is located on the Z and W chromosomes of the frog *Glandirana rugosa* and is expressed more highly in ZW than in ZZ gonads at the early tadpole stage. The expression of the W-borne allele is dominant and contributes to a higher expression in ZW gonads. The W-*Sox3* is expressed in somatic (mesonephric) cells around the boundary between the medulla and cortex of the ZW gonad. The disruption of *Sox3* produced sex-reversed ZW males and hermaphrodites, suggesting that Sox3 is a female determiner in this species. In contrast, the sex-determining function of the sex-linked androgen receptor gene proposed by Fujii et al. (2014) and Oike et al. (2017) [17,18] was rejected in our transgenic analysis. During the evolution of the ZZ-ZW system in the frog, another population of the XX-XY system sharing homologous sex chromosomes with the ZZ-ZW system evolved and exists in a different population. In future studies, we will identify the male-determining gene of the XX-XY system to elucidate the transition mechanism involved in male to female heterogametic sex determination.

## Figures and Tables

**Figure 1 biomolecules-14-01566-f001:**
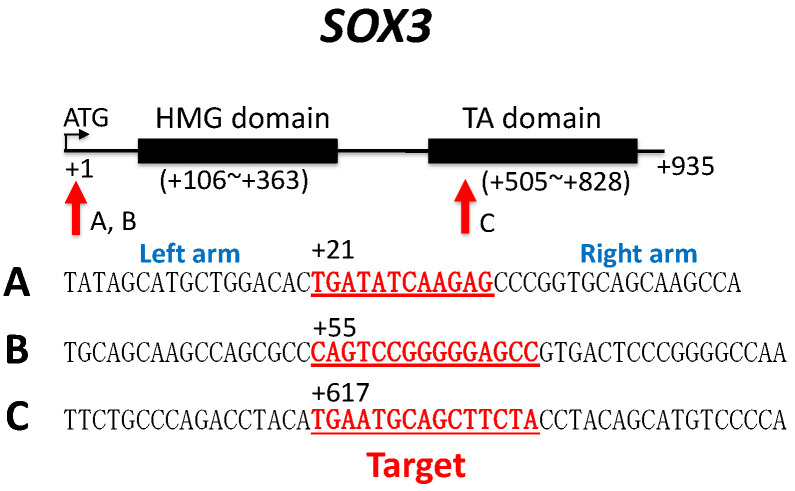
Target sequences of *G. rugosa Sox3* for TALEN knockout analysis. The red arrows indicate the locations of the target sequences of A, B and C, which are indicated in red and underlined.

**Figure 2 biomolecules-14-01566-f002:**
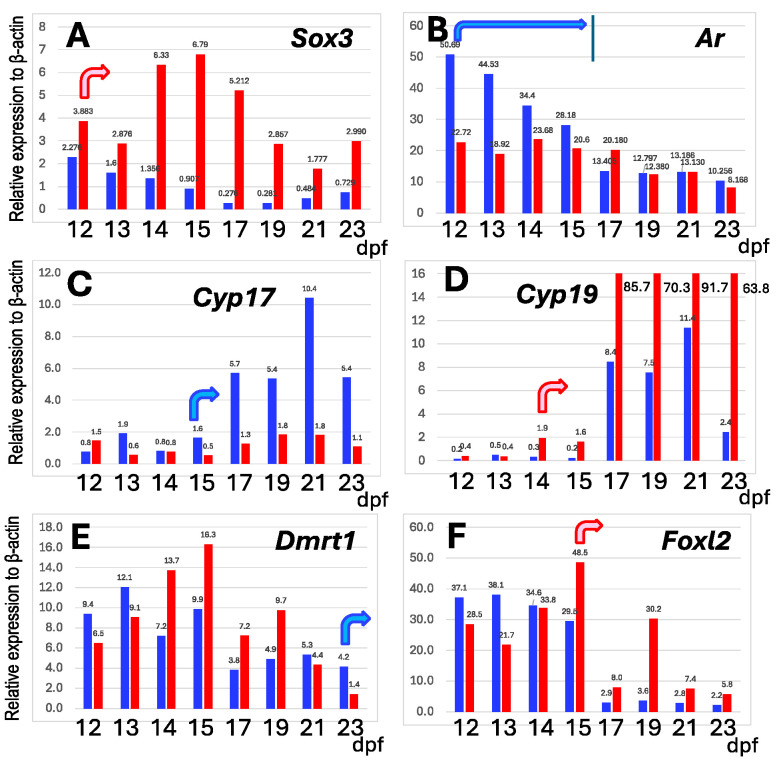
Expression profiles of six genes in gonad–mesonephros complexes of ZZ males (blue) and ZW females (red) of *G. rugosa* during early tadpole development. The expression rates of the genes are calculated using β-actin gene as the reference. Curved arrows indicate the onset stages of sexually dimorphic gene expressions, and vertical line in B indicates the end of the sexually dimorphic *Ar* expression. *Sox3* and *Ar* are sex-linked (**A**,**B**), whereas the others, *Cyp17*, *Cyp19*, *Dmrt1*, and *Foxl2 *(**C**–**F**), are autosomal genes. dpf, days post fertilization.

**Figure 3 biomolecules-14-01566-f003:**
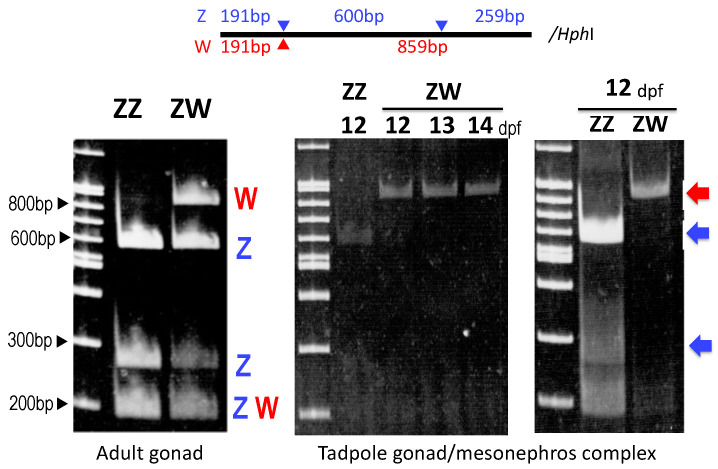
Allelic expression of *Sox3* gene. Amplified *Sox3* cDNA fragments were digested with *Hph*I. ZZ shows three bands because Z-*Sox3* cDNA has two *Hph*I sites (triangles in blue), whereas ZW shows four bands because W-*Sox3* has a single site (triangle in red). The **left panel** is *Sox3* cDNA amplified from adult gonads, and the **center** and **right panels** are those from the gonad–mesonephros complexes of early tadpoles. *W-Sox3* (red arrow) is dominantly expressed compared to *Z-Sox3* (blue arrows) in ZW tadpoles during 12–14 dpf. The polymerase chain reaction cycle numbers are 35 and 40 in the **left/middle** and **right panels**, respectively. The restriction map is shown on the top.

**Figure 4 biomolecules-14-01566-f004:**
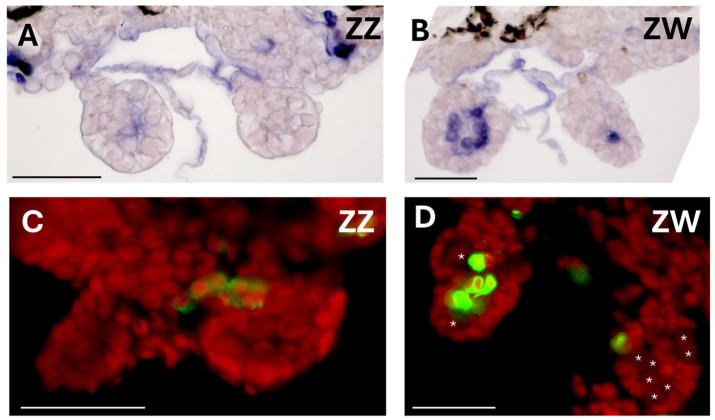
Localization of Sox3 protein in ZZ and ZW gonads of tadpoles at 18 dpf. Staining with alkaline phosphatase conjugated with the 2nd antibody and NBT (**A**,**B**) and with FITC conjugated with the 2nd antibody and PI (**C**,**D**). The signal of the anti-Sox3 antibody is blue in (**A**,**B**) and green in (**C**,**D**). Nuclei are stained red. Germ cells are indicated by asterisks. Bar, 50 μm.

**Figure 5 biomolecules-14-01566-f005:**
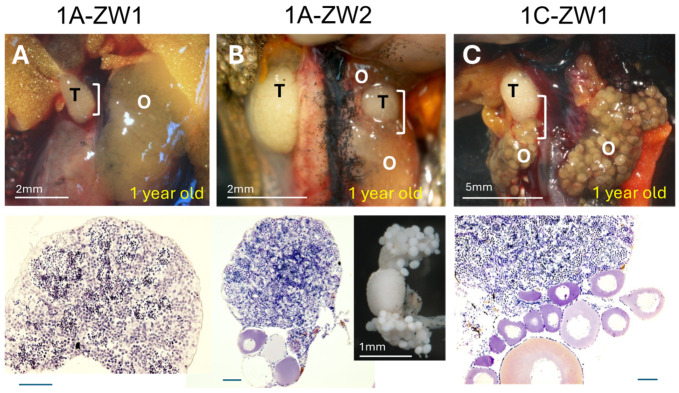
External morphology and inner structures of gonads of ZW hermaphrodites (**A**–**C**) produced by disrupting *G. rugosa Sox3*. Their ages are indicated at the bottom of the pictures in yellow. The gonad pictures are at the **top** and their histological sections stained with HE are in the **lower panel**. The sectioned regions of gonads are indicated using white vertical bars. T, testis and O, ovary. In (**B**), the fixed gonad with Nawashin is shown on the right side of the histological section. Bar, 100 μm (**lower panel**).

**Figure 6 biomolecules-14-01566-f006:**
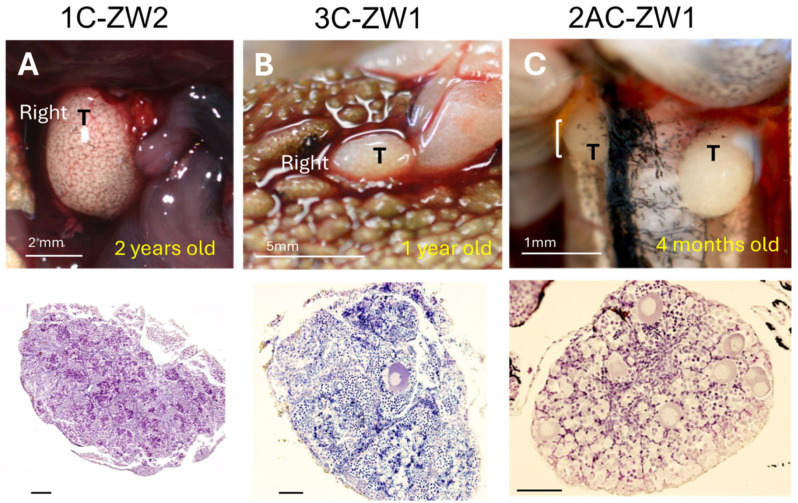
External morphology and inner structures of gonads of ZW sex-reversed males (**A**–**C**) produced by disrupting *G. rugosa Sox3*. Their ages are indicated at the bottom of the pictures in yellow. Gonad pictures are on the **top** and their histological sections stained with HE are in the **lower panel**. The sectioned region of the gonad is indicated using a white vertical bar. T, testis. “Right” means right testis. Bar, 100 μm (**lower panel**).

**Figure 7 biomolecules-14-01566-f007:**
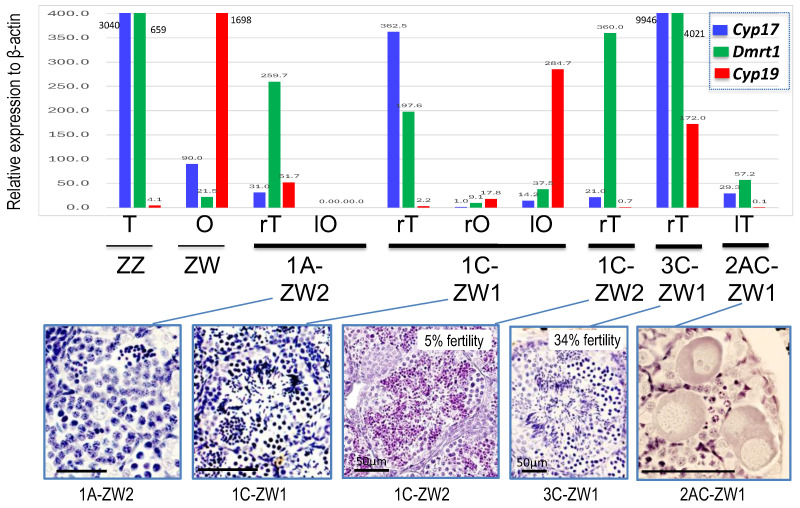
Expressions of male-specific genes of *Cyp17* and *Dmrt1* and a female-specific gene of *Cyp19* in the gonads of ZW sex-reversed males and hermaphrodites, as well as those of the control ZZ male and ZW female. The relative expression rates of the genes were calculated using β-actin as the reference gene. Magnified histological sections of the testes are shown in the **lower panel**. r, right; l, left; T, testis; O, ovary. In 1C-ZW2 and 3C-ZW1, their fertility rates relative to control males are shown on the top-right of lower panel (see Appendix A). Bars, 100 μm in 1A-ZW2, 1C-ZW1 and 2AC-ZW1.

**Figure 8 biomolecules-14-01566-f008:**
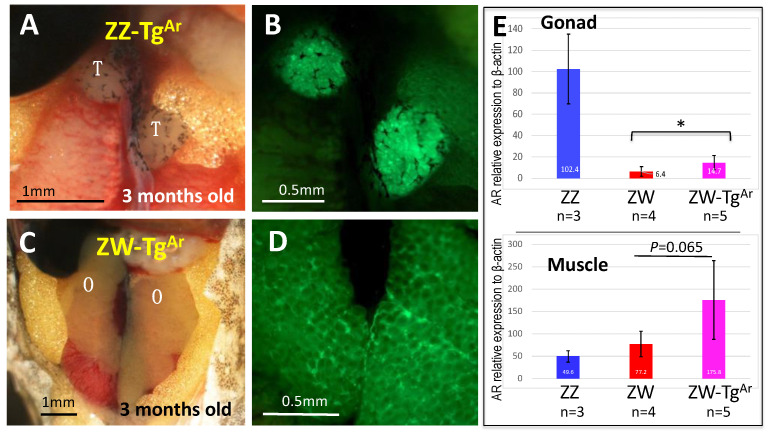
External morphology of gonads and expression of the androgen receptor (*Ar*) gene in the *Ar*-transgenic frogs. (**A**,**C**) The gonads of *Ar*-transgenic ZZ male and ZW female, respectively, and (**B**,**D**) the EGFP expression in their gonads. The expression of *Ar* in the gonads and muscles of the control ZZ, ZW, and transgenic ZW frogs are shown in (**E**). *, *p* < 0.05 (one-way ANOVA). Vertical bar, standard deviation. T, testis; O, ovary.

**Figure 9 biomolecules-14-01566-f009:**
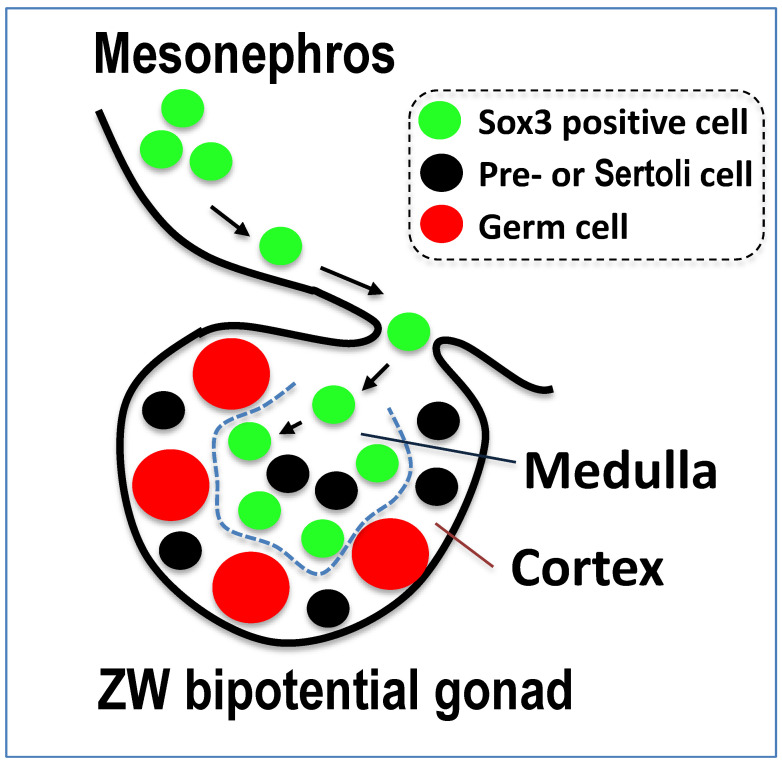
A hypothesis of ovary determination by the sex-linked *Sox3* gene in *G. rugosa*. Arrows indicate the movement of Sox3-positive cells. The dotted line indicates the boundary between the medulla and cortex.

**Table 1 biomolecules-14-01566-t001:** Sex reversal caused by the injection of TALEN vectors targeting the *Sox3* gene.

Crossing	Target Sequence	Feeding Tadpole	Metamorphosed Frog	Genetic Sex	Phenotypic Sex		
					Male	Female	Total
Na1 ^(1)^	A	25	15	ZZ	9	0	9
				ZW	2	4	6
Na2	B	35	19	ZZ	14	0	14
				ZW	0	5	5
Na1	C	36	26	ZZ	10	0	10
				ZW	2	14	16
Na3	C	39	20	ZZ	8	0	8
				ZW	1	11	12
Na2	A and C	19	19	ZZ	14	0	14
				ZW	1	4	5
Na4	B and C	24	14	ZZ	6	0	6
				ZW	0	8	8

^(1)^ Na, Nagaoka population (Niigata Prefecture) with ZZ-ZW-type sex chromosomes.

**Table 2 biomolecules-14-01566-t002:** Gonadal phenotype of sex-reversed ZW males and hermaphrodites.

Sex-Reversed ZW Males and Hermaphrodites	Gonad	
	Right	Left
1A-ZW1	Tests	Ovary
1A-ZW2	Testis	O-T-O ^(2)^
1C-ZW1	T-O ^(1)^	Ovary
1C-ZW2	Testis	Testis
3C-ZW1	Testis	Testis
2AC-ZW1	Testis	Testis

^(1)^ Posterior testis with anterior ovary; ^(2)^ A small testis within an ovary.

**Table 3 biomolecules-14-01566-t003:** Mutations identified in *Sox3* genes from the gonads of ZW sex-reversed males and hermaphrodites.

ZW Males and Hermaphrodites	Tissue DNA	No. of Clones Sequenced	Wild Type	Mutation (In-Frame)	Mutation Rate (%)
1A-ZW1	Right testis gDNA	28	15	13 (1)	46.4
	Left Ovary gDNA	23	9	14 (0)	60.9
1A-ZW2	Right testis cDNA	22	10	12 (1)	54.5
	Left Ovary cDNA	14	10	3 (1)	21.4
1C-ZW1	Right testis gDNA	18	2	16 (0)	88.9
	cDNA	17	1	16 (2)	94.1
	Right Ovary gDNA	24	4	20 (7)	79.2
	cDNA	13	2	11 (3)	84.6
	Left Ovary gDNA	19	6	13 (1)	68.4
1C-ZW2	NE				
3C-ZW1	Right testis gDNA	21	11	11 (0)	47.6
	cDNA	27	23	4 (4)	14.8
2AC-ZW1	Right test gDNA	16	6	10 (2)	62.5
	cDNA	13	5	8 (0)	61.5

gDNA, genomic DNA; cDNA, complementary DNA; NE, not examined.

## Data Availability

All data are presented in this paper.

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
