# Peer review of "Disruption of Sex-Linked Sox3 Causes ZW Female-to-Male Sex Reversal in the Japanese Frog Glandirana rugosa"

_biomolecules, 2024, doi:10.3390/biom14121566_

Round 1

Reviewer 1 Report

Comments and Suggestions for Authors

Minor comments on the manuscript

Lines: 220-223

Authors

“A positive signal was detected in the somatic cells of the gonads of ZZ males and ZW females at 18 dpf, and it was located at the medulla or at the boundary between the medulla and cortex in ZW females, whereas it was outside the gonads of ZZ 222 males (Figure 4).

Reviewer:

Fig. 4 does not clearly distinguish between somatic and germ cells, so I suggest deleting the term “somatic cells.”

Since your current results suggest that Sox3 mutations in the ZW testes are heavily expressed in somatic cells and slightly expressed in germ cells, it would be important to identify both kinds of cells directly in the developing gonads of G. rugosa.

Author Response

Dear reviewers,

Thank you for the comments. They are smart and excellent and are very helpful for improving our manuscript. Here, we respond to all the comments one by one, written in blue. The revised or added parts in the text are written in red.

Sincerely,

Ikuo Miura

Reviewer 1

Minor comments on the manuscript

Lines: 220-223

Authors

“A positive signal was detected in the somatic cells of the gonads of ZZ males and ZW females at 18 dpf, and it was located at the medulla or at the boundary between the medulla and cortex in ZW females, whereas it was outside the gonads of ZZ males (Figure 4).

Reviewer:

Fig. 4 does not clearly distinguish between somatic and germ cells, so I suggest deleting the term “somatic cells.” 

Since your current results suggest that Sox3 mutations in the ZW testes are heavily expressed in somatic cells and slightly expressed in germ cells, it would be important to identify both kinds of cells directly in the developing gonads of G. rugosa.

The germ cells during the early developmental stages are large in size because their cytoplasm is larger than that of somatic cells and are stained faintly with PI. In addition, the Sox3 positive cells have the cellular origins from mesonephros. In fact, we captured a figure showing the movement of the Sox3 positive cells coming from mesonephros into gonad. Then, we added one supplementary figure S3 to show the cell movement, and also asterisks indicating germ cells are added to Figure 4 as in Figure S3, where the Sox3 expressing cells are not overlapped with germ cells. Therefore, we concluded that the Sox3 positive cells belong to one lineage of the somatic cells (mesonephric cells), but not germ cells.

Explanation about movement of Sox3 positive cells is added in L254-258:

“In addition, we observed that the Sox3 positive cells were migrating from mesonephros into ZW bipotential gonad (Figure S3).“  

Reviewer 2 Report

Comments and Suggestions for Authors

The objective of this paper was to determine the effect of disruption of sex-linked Sox3 in terms of sex determination in the Japanese frog Glandirana rugosa.    The authors observed that that the levels of expression of Sox3, a known maleness gene, in the gonads/mesonephroi were much higher in ZW females than in ZZ males.    The authors used q-PCR and found significantly higher expression of Sox3 in the gonad of females than in males.    The results for SOX3 is significant and clear, but q-PCR results for several other sex determination-related genes are mixed.    For example, Dmrt1 is a maleness gene, but was found to be expressed at higher levels in females 14-19 dpf.    Such mixed results need to be thoroughly explained as to why and how they are related to sex determination.
The allelic expression of Sox3 is very interesting, but need to be clearly presented as to why and how that is allelically expressed, and do that have a biological significance.
Spatial expression was addressed by immunohistology, which is convincing.
The knockout experiments using TALEN is conducted to address the function of SOX3 in sex determination.    Sex reversal in knockout individuals suggest the involvement of SOX three in the sex determination cascade, but by no means mean SOX3 is the sex determination gene.    First, SOX3 is an upstream transcription factor, its knockout is expected to have an effect in sex phenotype.    However, how a maleness gene now function as a femaleness gene in the Japanese frog must be explained.
The analysis of expression of several key genes in the knockout is very nice.
Section 3.5 is problematic.    Up-regulation Ar gene expression is only one step.    If Ar gene were the master sex determination gene, the whole system has evolved such that androgen hormones are lacking.    Even if Ar receptor gene is upregulated, without the ligand, it would be difficult to conclude the function of Ar.
In the Discussion: The following sentences are highly problematic:
“Such non-coding RNA might have been borne around the W-borne Sox3 region in the frog and changed the original expression profile of sex-linked Sox3.    Or, some regulatory connection between Sox3 and Sox2 can exist in the sex-determining cascade, as has been shown in neurogenesis [36].”    Once again, the authors should provide an convincing hypothesis as to how SOX3 determines femaleness.    If it functions like DM-W in African frog, through dominant negative regulation, the evidence must be shown to suggest that the W copy SOX3 is a truncated form or some other mutations that exist within the gene on W. Otherwise, the whole paper reported lots of interesting observations, but no interpretation.

Minor changes should be made, especially in the usage of English.    I have the following specific suggestions:

Title should be changed, because it is confusing now, to something like:

Disruption of sex-linked Sox3 causes sex-reversal of ZW to phenotypic male in  the Japanese frog Glandirana rugosa

Figure legends of Figure 2, the color code should be explained.    I assume the red is ZW female, but it should be clear.

Similarly, subtitle of 3.3 should be changed to:

3.3.    Disruption of Sox3 causes sex-reversal of genotypic ZW to phenotypic males

subtitle of 3.5 should be changed to:

3.5.    The sex-linked androgen receptor gene Ar is not up-regulated in sex-reversed neomales

Overall, the work is interesting and relevant to the field and to the readers of the journal, but it lacks a sound hypothesis as to why and how a maleness gene function as a femaleness determinant in the Japanese frog.

Author Response

Dear reviewers,

Thank you for the comments. They are smart and excellent and are very helpful for improving our manuscript. Here, we respond to all the comments one by one, written in blue. The revised or added parts in the text are written in red.

Sincerely,

Ikuo Miura

Reviewer 2

The objective of this paper was to determine the effect of disruption of sex-linked Sox3 in terms of sex determination in the Japanese frog Glandirana rugosa.    The authors observed that that the levels of expression of Sox3, a known maleness gene, in the gonads/mesonephroi were much higher in ZW females than in ZZ males.    The authors used q-PCR and found significantly higher expression of Sox3 in the gonad of females than in males.    The results for SOX3 is significant and clear, but q-PCR results for several other sex determination-related genes are mixed.    For example, Dmrt1 is a maleness gene, but was found to be expressed at higher levels in females 14-19 dpf.    Such mixed results need to be thoroughly explained as to why and how they are related to sex determination.

Cyp 17 and Cyp19 are involved in syntheses of testosterone and estradiol, respectively, and Dmrt1 and Foxl2 are involved in testicular and ovary differentiation, respectively, in vertebrates. Their expressions are much higher in male or female gonads, but it is not all or none. These hold true for frogs and some papers show their sex specific expressions and functions in frogs. Therefore, they are called, sex-differentiation genes. In fact, in our present study, the male specific or female specific gene expression begins at the fixed stage (after under control of sex determining gene) during early tadpole development and is kept through life in gonads of adults. However, before the fixed stage, their expressions show swinging between sexes, probably because they are not yet under the control of the sex-determining gene and reflect the individual variations because we used ten male and ten female tadpoles for q-PCR analysis.

According to the reviewer’s comment, we added the following sentences in L212-228 in the text:

“Cyp17 and Cyp19 are involved in syntheses of testosterone and estradiol, respectively, and Dmrt1 and Foxl2 are involved in testis and ovary differentiation of vertebrates, respectively. At early tadpole stages, the expressions of the four autosomal genes swung between males and females, probably due to individual variations (ten males and ten females were used for one stage analysis), but after the fixed stages their sexually dimorphic expressions became stable, which are designated here the onset stage of sexually dimorphic expression.”

The allelic expression of Sox3 is very interesting, but need to be clearly presented as to why and how that is allelically expressed, and do that have a biological significance.

The acquisition of the unique expression of W-born Sox3 is crucially important  for evolution of the female determining gene and ZZ-ZW system in the frog. We speculate that the Sox3 gene on the W chromosome may have acquired an enhancer element to induce its higher and ectopic expression in the somatic cells of gonad during evolution of the original hybrid-population between the West-Japan and G. reliquia (Miura et al., 2022).

We added the above explanation in the discussion section, L353-357:

“ Acquisition of the unique expression of W-born Sox3 may have been crucial for birth of the female determining gene and ZZ-ZW system in the frog. We hypothesize that the Sox3 gene on the original chromosome 7 acquired an enhancer element to induce its higher and ectopic expression in the somatic cells of gonad during evolution of the original population.”

And, we deleted the following sentences from the discussion section 4.1 (L353) because overlapped with 4.3 section: “We speculate that somatic cells around the boundary between the medulla and cortex play a role in ovary determination by inhibiting germ cell migration or movement from the cortex to the medulla and by forming the ovarian cavity. The Sox3 expressing somatic cells in the frog may differ in lineage from the Sertoli cells in vertebrates to induce testis determination.”

Spatial expression was addressed by immunohistology, which is convincing.

Thanks.

The knockout experiments using TALEN is conducted to address the function of SOX3 in sex determination.    Sex reversal in knockout individuals suggest the involvement of SOX 3 in the sex determination cascade, but by no means mean SOX3 is the sex determination gene.    First, SOX3 is an upstream transcription factor, its knockout is expected to have an effect in sex phenotype.    However, how a maleness gene now function as a femaleness gene in the Japanese frog must be explained.

This is a very important question. We added the explanation in the new discussion section in L376-399:

“4.3. How does Sox3 determine an ovary differentiation in ZW bipotential gonad?

The male determining gene Sry of eutherians, which evolved from Sox3, is first expressed in somatic (Sertoli) cells of XY bipotential gonad and determines its fate toward testis. Even the homologous Sox3 on the X chromosome, if ectopically expressed in the Sertoli cells, can determine testis differentiation [36]. At the next stage of the testis differentiation, the mesonephric (epithelial) cells migrate into the XY gonad to form testicular cords [37,38]. Then, how does the W-Sox3, which is similarly expressed in somatic cells of bipotential gonad, determine an ovary in the frog G. rugosa? The function of Sox3 is completely opposite to the male determination in eutherians and medaka fish. Our hypothesis is that W-Sox3 is first expressed highly in mesonephros and then migrate into ZW bipotential gonad, taking the position at center or boundary between the medulla and cortex (Figs. 4 and 9; Figure S3). The Sox3 expressing mesonephric cells separate the medulla from cortex to inhibit movement of germ cells from the cortex and form an ovarian cavity. Together, W-Sox3 may promote up-regulation of Cyp19 in the cells, as suggested by Oshima et al. (2009) [26], to synthesize estradiol and contribute to the ovary development and growth. Hereafter, it is necessary to identify the direct target gene and the downstream genes of Sox3 to confirm our hypothesis.”

The analysis of expression of several key genes in the knockout is very nice.
Section 3.5 is problematic.    Up-regulation Ar gene expression is only one step.    If Ar gene were the master sex determination gene, the whole system has evolved such that androgen hormones are lacking.    Even if Ar receptor gene is upregulated, without the ligand, it would be difficult to conclude the function of Ar.

Yes, we agree. Therefore, our conclusion is that Ar is not involved in sex determination in the frog G. rugosa . We proved it based on the results of our transgenesis: just upregulation of Ar does not change the phenotypic sex of ZW female gonad. We rejected the Ar sex determination theory proposed by Fujii et al. and Oike et al.. It is discussed in the section 4.4.     .

In the Discussion: The following sentences are highly problematic:
“Such non-coding RNA might have been borne around the W-borne Sox3 region in the frog and changed the original expression profile of sex-linked Sox3.    Or, some regulatory connection between Sox3 and Sox2 can exist in the sex-determining cascade, as has been shown in neurogenesis [36].”    Once again, the authors should provide an convincing hypothesis as to how SOX3 determines femaleness.    If it functions like DM-W in African frog, through dominant negative regulation, the evidence must be shown to suggest that the W copy SOX3 is a truncated form or some other mutations that exist within the gene on W. Otherwise, the whole paper reported lots of interesting observations, but no interpretation.

Thank you for this comment. We added our interpretation in the new discussion section 4.3 by responding to your previous comment.

Minor changes should be made, especially in the usage of English.    I have the following specific suggestions:

Title should be changed, because it is confusing now, to something like:

Disruption of sex-linked Sox3 causes sex-reversal of ZW to phenotypic male in  the Japanese frog Glandirana rugosa

We are sure that our title is not wrong, because “XX male sex-reversal” is often used in the papers describing sex-reversal of mouse and humans. Probably, because people are not familiar to the ZW system, our title looks confusing. Therefore, we changed our title to:

“Disruption of sex-linked Sox3 causes ZW female-to-male sex reversal in the Japanese frog Glandirana rugosa”.

Figure legends of Figure 2, the color code should be explained.    I assume the red is ZW female, but it should be clear.

Thanks. We missed it, and added the explanation in Figure 2 legend.

Similarly, subtitle of 3.3 should be changed to:

3.3.    Disruption of Sox3 causes sex-reversal of genotypic ZW to phenotypic males

According to the title, we revised to:

3.3.  Disruption of Sox3 causes ZW female-to-male sex reversal

subtitle of 3.5 should be changed to:

3.5.    The sex-linked androgen receptor gene Ar is not up-regulated in sex-reversed neomales

Based on our results, we changed the title as follows:

3.5. ZW female is not sex-reversed by up-regulating sex-linked androgen receptor gene”

Overall, the work is interesting and relevant to the field and to the readers of the journal, but it lacks a sound hypothesis as to why and how a maleness gene function as a femaleness determinant in the Japanese frog.

Thank you very much for the comment. We discussed it in the discussion section 4.3.

Round 2

Reviewer 2 Report

Comments and Suggestions for Authors

My concerns are mostly addressed.